# Probing the composition dependence of residual stress distribution in tungsten-titanium nanocrystalline thin films

Rahulkumar Jagdishbhai Sinojiya [1], Priya Paulachan[1], Fereshteh Falah Chamasemani[1], Rishi Bodlos [1], René Hammer[1], Jakub Zálešák[2], Michael Reisinger[3], Daniel Scheiber [1], Jozef Keckes[2], Lorenz Romaner[2] & Roland Brunner [1✉]

Nanocrystalline alloy thin films offer a variety of attractive properties, such as high hardness, strength and wear resistance. A disadvantage is the large residual stresses that result from their fabrication by deposition, and subsequent susceptibility to defects. Here, we use experimental and modelling methods to understand the impact of minority element concentration on residual stresses that emerge after deposition in a tungsten-titanium film with different titanium concentrations. We perform local residual stress measurements using micro-cantilever samples and employ machine learning for data extraction and stress prediction. The results are correlated with accompanying microstructure and elemental analysis as well as atomistic modelling. We discuss how titanium enrichment significantly affects the stress stored in the nanocrystalline thin film. These findings may be useful for designing stable nanocrystalline thin films.

---

[1] Materials Center Leoben Forschung GmbH, A-8700 Leoben, Austria. [2] Department of Materials Science, Montanuniversität Leoben, A-8700 Leoben, Austria. [3] KAI Kompetenzzentrum Automobil- und Industrieelektronik GmbH, A-9524 Villach, Austria. ✉email: roland.brunner@mcl.at

 1

Based on their characteristic internal length scales, nanocrystalline metallic alloy thin films provide a variety of interesting and useful properties[1–6]. Due to their enhanced functionality, such thin films are highly interesting for engineering applications. However, a big drawback concerns their generation of residual stresses evolving during deposition and their subsequent susceptibility to defects[7–9], which are limiting the performance of the functional film after deposition. Therefore, the perception of the residual stresses is crucial. Here, one can observe the stresses within the thin film during deposition in-situ[6,10,11] or ex-situ after the deposition is concluded[5,12,13]. The latter provides the potential to witness the stresses after curing or certain loading conditions[12–15]. Chason et al.[2,8,11,16] developed a kinetic model and used real-time wafer curvature experiments to understand the residual stress evolution during deposition. In general, methods such as wafer curvature[1,17,18], X-ray diffraction[14], and Raman spectroscopy[15] have been developed to determine the global residual stresses in thin films. However, not only the knowledge about the averaged stress of the film but also the experimental evidence about the position of the maximum stress in relation to the interfaces, the overall stress distribution as well as the local assignment of tensile and compressive stress are crucial for the long-life cycle behaviour of the deposited film. Hence, methods enabling the accurate local measurement of the residual stresses over the film thickness[12,13,19,20] are essential. In particular, methods which are material and surface independent or utilized in a lab environment[12,13,19] as well as enabling possibilities for automated acquisition, are in favour. Machine learning (ML)-driven approaches foster extended possibilities with respect to automated acquisition and analysis[21–23]. Recently, remarkable results have been shown for instance with data-based prediction models[21–24].

In general, the material microstructure influences the material characteristics and a profound knowledge of the underlying structure-property relationships is crucial for an improved understanding as shown in e.g.[21,22,25]. The microstructure of the thin films cannot be neglected to understand the material behaviour[26–29]. For nanocrystalline systems in particular, the consideration of the grain structure becomes relevant[30,31]. A possible approach to stabilize the nanocrystalline structure is by alloying with a minority element[30,31]. In this context tungsten-titanium ($W_{1-x}Ti_x$) thin films, display an interesting system, not only from the scientific but also from the application point of view e.g. as for diffusion barriers in the semiconductor industry[32,33]. The frequent preference of a minority element to segregate at the grain boundary, rather than to be dissolved in the interior of the grain, impacts the material characteristics[1,2,7–9,30–32]. In past, Boyce and co-workers developed a model which evaluates the impacts of the grain boundary segregation of the solute during the grain growth process[34]. Previous attempts to explain segregation behaviour in $W_{1-x}Ti_x$ have been made on the basis of an idealized thermodynamic description of the W-Ti alloy[30,31,35,36]. However, recent atomistic calculations of the mixing enthalpy of W-Ti[37] differ qualitatively from the previously assumed shape within the regular nanocrystalline solution model. Furthermore, also the segregation energy of Ti in W, as obtained from the Miedema model[30] is in qualitative disagreement with results from ab-initio simulations[38–41]. Therefore, the mechanisms behind stable nanocrystalline thin films are still obscure. Such further insights are crucial to fabricate stable nanocrystalline thin films suitable for engineering applications.

In this paper, we have developed a unique framework conducting experimental and modelling methods as well as machine learning (ML) assisted analysis to gain a crucial understanding of how Ti, as the minority element, impacts the residual stress stored in a fully functional deposited nanocrystalline W-Ti thin film. We utilize the ion layer removal (ILR) method by fabricating micro cantilevers with different at% Ti. We modify the deflection of the cantilever by material removal on the micro cantilever in the so called ILR-zone. A general regression neural network (GRNN) is developed to extract the deflection automatically from the generated scanning electron microscopy image data stacks for each material removal step in the ILR-zone. The deflection is used in the finite element method (FEM) modelling to calculate the residual stress distribution along the thickness of the film, with a resolution of 15 nm. Further, a ML-based feed-forward model, constructed by two multivariable regression architectures is developed to predict the residual stress crossing between the compressive to the tensile regime with respect to minority element concentration. The obtained stress distribution on nm-scales, is correlated with microstructure analysis accompanied with elemental analysis. Furthermore, we link the findings with density functional theory to inquire why the observed Ti enrichment can be strongly reduced at smaller concentrations and discuss how it significantly affects the stress configuration stored in the nanocrystalline thin film. For the first time, we provide a thermodynamic treatment of both the bulk and the grain boundary of $W_{1-x}Ti_x$ with ab-initio calculations.

## Results

**Microstructure characterization of the nanocrystalline $W_{1-x}Ti_x$ thin films.** The investigated $W_{1-x}Ti_x$ thin film stacks consists of a silicon substrate Si (100), a 100 nm non-stoichiometric thick silicon oxide (SiO) layer, and the $W_{1-x}Ti_x$ alloy thin film with a thickness of about 280 nm. Three different $W_{1-x}Ti_x$ compositions are used with x = 15, 20 and 30 at%. The thin films are deposited on the Si (100) substrate by physical vapour deposition (PVD). The deposition of the $W_{1-x}Ti_x$ layer is deposited analogous to[42,43]. The deposition is carried out using the same target and the Ti concentration is varied by adjusting the sputter pressure and the bias voltage. The process pressure during PVD is adjusted by a constant argon (Ar) gas flow, used as discharge gas. A ~280 nm thick W-Ti layer is deposited with a constant deposition rate. The deposition temperature, the distance to the target and the sputter power are kept constant for all three compositions.

We conduct field emission scanning electron microscopy (FESEM) in combination with electron back scattered diffraction (EBSD) as well as field emission scanning electron microscopy transmission Kikuchi diffraction (FESEM-TKD) to gain substantial information about the microstructure of the different $W_{1-x}Ti_x$ configurations on different length scales. Further, by connecting the underlying microstructure with the locally emerging residual stresses over the nanocrystalline film thickness a profound knowledge of the structure-property relationships shall be collected. Secondary electron (SE)-FESEM images in Fig. 1a indicate on the $W_{1-x}Ti_x$ surface (x-y-plane) a lamella-like microstructure for all three samples, which depends on the concentration of Ti. A similar surface-related microstructure for $W_{1-x}Ti_x$ thin films has been observed previously[32]. We segment the lamella-like microstructure in the FESEM image data for the three different $W_{1-x}Ti_x$ thin film configurations to quantify the lamella density and orientation, see Fig. 1a. Further details about the segmentation are presented in Supplementary Note 1. The segmented lamellas are coloured according to their orientation from 0° to 90°. The evaluated lamella density is 1640 µm⁻², 1449 µm⁻² and 730 µm⁻² for 15, 20 and 30 at% Ti, respectively. Clearly, for 30 at% Ti, a less dense lamella structure with thicker lamellas is depicted for the same region of interest (ROI). The orientation analysis suggests that more lamellas are oriented along 45° for 15 and 20 at% than for 30 at%. For 30 at% Ti a rather inhomogeneous lamellas distribution is depicted.

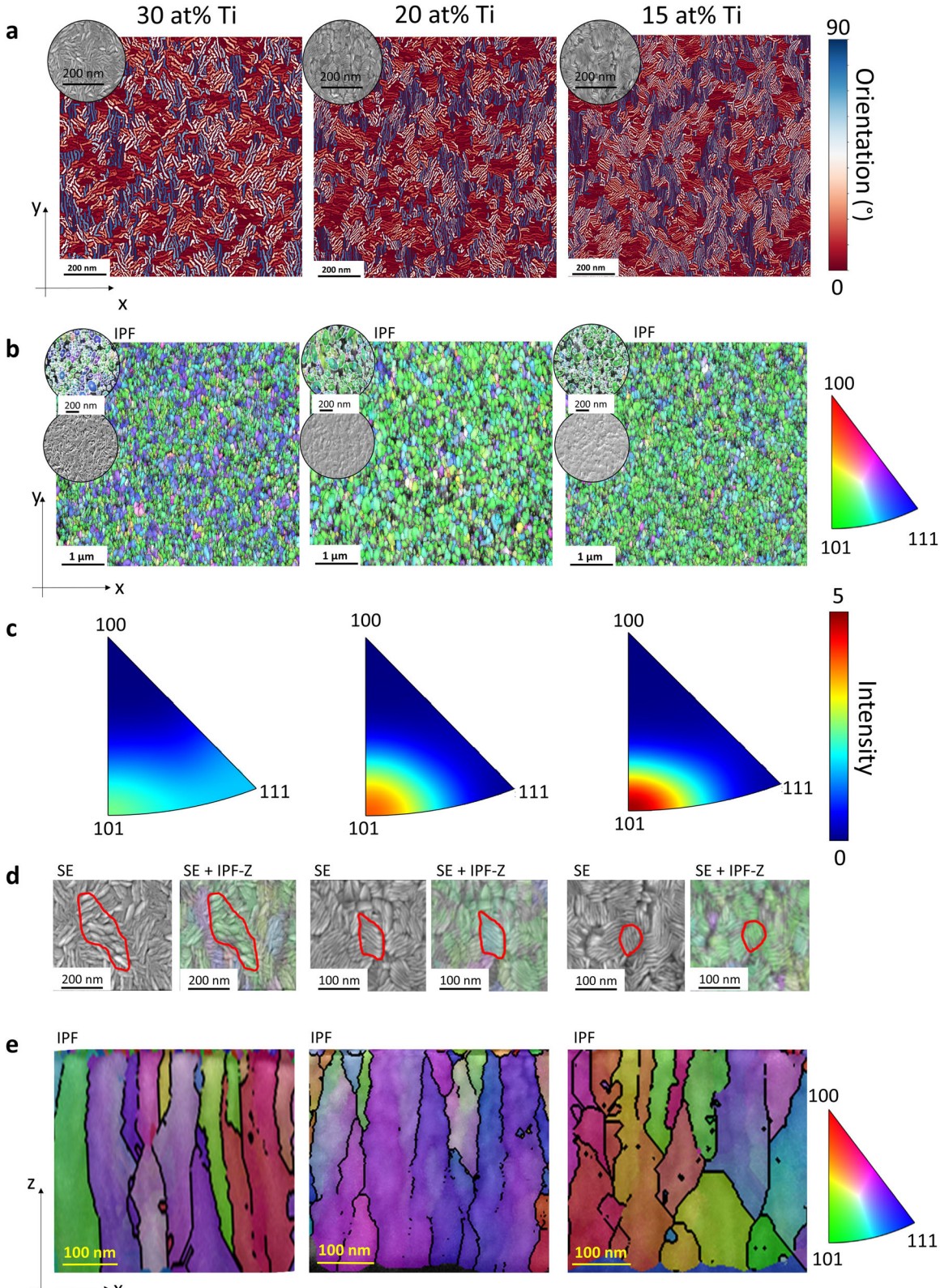

Figure 1b shows the FESEM-EBSD inverse pole figure (IPF)-Z data from the surface (x-y-plane), for a region of interest (ROI) with 5.8 × 5.2 μm². The insets illustrate the corresponding secondary electron (SE)-FESEM images. The extracted mean diameter and corresponding standard deviation (see method section) for 15, 20 and 30 at-% Ti is 117 nm ± 58 nm, 125 ± 59 nm and 108 nm and ± 43 nm, respectively.

In Fig. 1c we plot the intensity distribution extracted from the given IPF-Z to visualise the crystallographic orientation for the different $W_{1-x} Ti_x$ thin films (x-y plane). For 30 at% Ti, the orientation is distributed between [111] and [101]. For 15 and 20 at% Ti, the pole figures reveal a maximum in the intensity at the [101] grain orientation. The misorientation map shown in the supplementary (see Supplementary Fig. 1) supports the results of

**Fig. 1 Microstructural analysis for 30 at% Ti, 20 at% Ti and 15 at% Ti, from left to right. a** Image analysed, SE-FESEM surface data for 30 at% Ti, 20 at% Ti and 15 at% Ti, respectively. The insets illustrate examples of grey value images at the given position. The colour bar indicates the modulus of the orientation angle of the lamellas, from 0 (red) to 90° (blue). **b** EBSD based IPF-Z for 30 at% Ti, 20 at% Ti and 15 at% Ti, respectively, representing the crystallographic orientation of the $W_{1-x}Ti_x$ thin films. Insets show the fitted ellipsoids for the grains and the corresponding SE-FESEM images, respectively. **c** EBSD based IPF-Z intensity distribution for the surface data to illustrate the crystallographic orientation for 30 at% Ti, 20 at% Ti and 15 at% Ti, respectively. **d** Exemplary region of interest indicate the projected FESEM greyscale image on top of the surface FESEM-EBSD data for 30, 20 and 15 at% Ti. Red line highlights for a single grain the correspondence between the grain and lamella orientation, depicted from FESEM-EBSD and FESEM, respectively. Within one grain, various lamellas are oriented along a certain angle. **e** Field emission scanning electron microscopy transmission Kikuchi diffraction (FESEM-TKD) for 30 at% Ti, 20 at% Ti and 15 at% Ti, respectively.

**Table 1 FESEM-EBSD surface (x-y plane) and FESEM-TKD cross-section (x-z) grain analysis illustrating the mean value and standard deviation for the minor axis, aspect ratio and mean area of the fitted ellipses as well as the grain density.**

|  | W-Ti | Minor axis of ellipse | Minor axis of ellipse | Aspect ratio of fitted ellipse | Aspect ratio of fitted ellipse | Grain Density | Area of ellipse |
|---|---|---|---|---|---|---|---|
|  | at% Ti | nm | nm |  |  | $\mu m^{-2}$ | $\times 10^3$ nm$^2$ |
|  |  | Mean | Standard deviation | Mean | Standard deviation |  | Mean |
| Surface | 15 | 58 | 29 | 1.86 | 0.62 | 105 | 4.6 |
|  | 20 | 60 | 30 | 1.94 | 0.62 | 83 | 5.2 |
|  | 30 | 50 | 22 | 2.07 | 0.70 | 150 | 3.9 |
| TKD Cross-section | 15 | 31 | 56 | 2.68 | 1.16 | 178 | 8.1 |
|  | 20 | 34 | 46 | 2.92 | 1.98 | 120 | 11.1 |
|  | 30 | 27 | 25 | 3.37 | 1.17 | 252 | 5.2 |

the pole figures. Here, it is clearly demonstrated that the misorientation is highest for the Ti concentration with 30 at%. Figure 1d provides further information about the correlation between the observed lamella and the grain microstructure obtained from the FESEM and FESEM-EBSD measurements, respectively. Hence, we project the FESEM data on the FESEM-EBSD images. Figure 1d suggests that each single grain consists of a certain number of lamellas oriented along a given direction. The FESEM images of the surface with different magnifications is shown in Supplementary Fig. 2.

Figure 1e, shows the cross-section (x-z plane) using FESEM-TKD to resolve the nanocrystalline grain structure of the investigated thin films for an ROI of $500 \times 300$ nm$^2$. We evaluate the grain shape for the surface and cross-section data by fitting the grains with ellipsoids (see Supplementary Fig. 3. and Supplementary Fig. 4). The corresponding statistics with respect to the minor axis, aspect ratio and area of the fitted ellipses as well as the grain density for the surface (x-y plane) and FESEM-TKD cross-section (x-z plane) is summarized in Table 1. The mean aspect ratio for the cross-section and surface is increasing with increasing Ti content. A stronger increase is demonstrated for the cross-sectional data.

According to Table 1, the 30 at% Ti-alloy shows the highest grain density with about 252 µm$^{-2}$ (150 µm$^{-2}$) for the cross-section (surface), respectively. Here, the mean area of the fitted ellipse shows the smallest value for the cross-section (surface) with $5.2 \times 10^3$ nm$^2$ ($3.9 \times 10^3$ nm$^2$). This is illustrated by the thin (in x-direction) and densely packed columnar grain structure observed in Fig. 1e.

The alloy with 20 at% Ti shows the lowest grain density with about 120 µm$^{-2}$ (83 µm$^{-2}$) for the cross-section (surface). Here, in comparison to the 30 at% Ti-alloy, a columnar grain structure with thicker (in x-direction) grains is observed (Fig. 1e). The extracted mean area (Table 1) of the fitted ellipse provides the largest measure for the cross-section (surface), with about $11.1 \times 10^3$ nm$^2$ ($5.2 \times 10^3$ nm$^2$).

The grain density for the cross-section (surface) of the 15 at% Ti sample (Fig. 1e), is smaller than for the 30 at% Ti with about 178 µm$^{-2}$ (105 µm$^{-2}$). The mean area (Table 1) of the fitted ellipses is for the cross-section (surface) $8.1 \times 10^3$ nm$^2$ ($4.6 \times 10^3$ nm$^2$). That

is, it lies between the 30 and 20 at% Ti alloy. As shown in the EBSD cross-sectional data (Fig. 1e) a rather less distinctive columnar structure is shown for 15 at% Ti compared to 20 and 30 at% Ti. Note, that the statistical analysis of the cross-sectional data for a larger ROI with 3.26 µm × 300 nm and measured by FESEM-EBSD in reflection, indicates a similar trend (see Supplementary Table 1).

**General regression neural network (GRNN) for an accurate determination of the micro-cantilever deflection.** We utilize the ion layer removal (ILR) method[12,13,19,20,44] to extract the local residual stress along the thickness of the nano-crystalline thin film layer. Therefore, we manufacture from the deposited films micron-sized cantilevers for each Ti at% and characterize in a first step the deflection of the free-standing beam, see Fig. 2. The micro-cantilevers as shown in Fig. 2a, are prepared by ion-slicing and finished by scanning electron-focused ion beam microscopy (SEM-FIB), see method section. The collected deflection data for each micro-cantilever, see Fig. 2a, is then further used as an input for the finite element modelling (FEM) to extract the corresponding residual stresses along the thin film thickness. Within a defined area we remove material from the micro-cantilever inside the SEM-FIB. The ion beam provides a cutting-step size of 15 nm. The defined area is also referred to the so-called ion layer removal- (ILR-) zone (Fig. 2a). The resulting deflection of the cantilever is imaged in-operando and the image data for each material-removal step and the associated deflection is collected and stored. The deflection change, can be detected, witnessing the position of an imprinted marker (line) located at the end of the cantilever (moving part) and comparing it's relative change to an additional marker (line) close-by, on the non-moving part. The subsequent FEM demands an accurate image analysis of the micro-cantilever deflection. Furthermore, an amount of up to about 61 SEM images per ILR cycle needs to be efficiently processed for a thickness of about 915 nm, including the $W_{1-x}Ti_x$ film and the Si and SiO with 635 nm. The cutting in the Si is used as a validation of the result. To tackle the need of the accuracy and efficiency, we apply a general regression neural network (GRNN)[45] designed to extract the deflection from the collected SEM image stacks (Fig. 2a). As an input for the GRNN we utilize

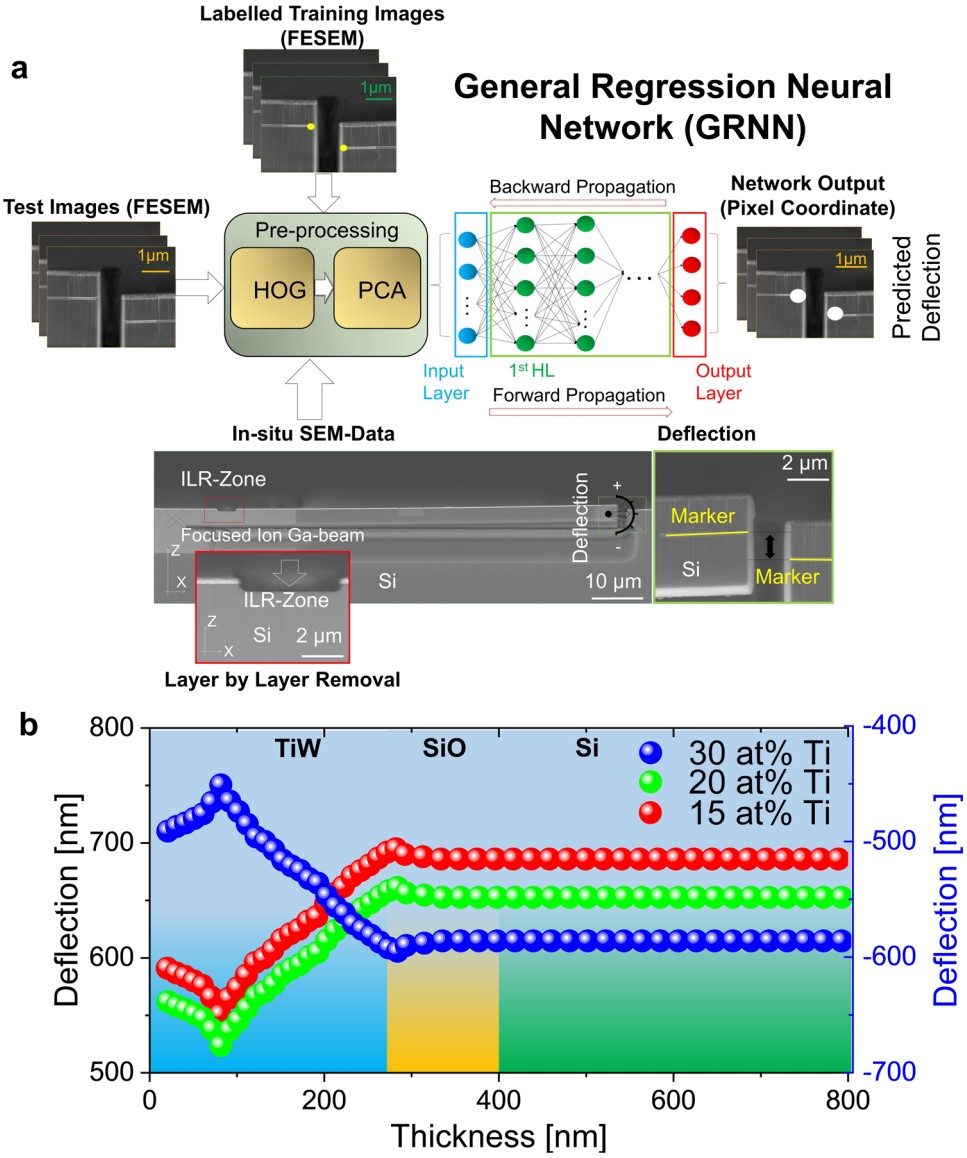

**Fig. 2 ML-assisted workflow to extract the deflection from the SEM-FIB image data for different removal steps. a** The schematic shows the ML-assisted workflow used to extract the deflection for different removal steps from the collected image data. The workflow includes: (1) the post-processing of the in-operando SEM-FIB image data utilizing histogram of gradient (HOG) and principal component analysis (PCA), (2) the training of the general regression neural network (GRNN) with the manually labelled image data, (3) the input of the GRNN using the post-processed image SEM-FIB data, and (4) the output generating the deflection. Two imprinted markers (highlighted with white points), one localized on the cantilever (moving part) and one located on the opposite site of the cantilever (non-moving part) are used to determine the deflection of the cantilever for different cutting depths. **b** Depicted deflection from the GRNN-model for 15 at% (red), 20 at% (green) and 30 at% (blue) Ti over the layer thickness with a step-size of 15 nm. Background indicates the different layers in the thin film stack: $W_{1-x}Ti_x$ (blue), SiO (orange), Si (green). The deflection profile changes significantly within the different $W_{1-x}Ti_x$ layer configurations. A constant deflection is shown for the SiO and Si layers. The latter makes sense since this layer should be stress-free.

post-processed SEM-FIB data. For the post-processing, histogram of gradient (HOG) feature extraction is applied to gain an accurate object detection in images with $15 \times 15$ pixels. Subsequently, principal component analysis (PCA) is performed to transform the features in smaller sub-sets, suitable to be used as an input for the GRNN, see Fig. 2a. In total, 400 features are extracted for the GRNN[45] (see method section).

The GRNN model is trained by using overall 600 manually labelled images, see Fig. 2a. Here, we label within the images the pixels associated with the markers used for the deflection evaluation. The root-mean-square error (RMSE) provides for the developed model an inaccuracy of about 2 nm. For a conventional image analysis (Supplementary Note 2 and Supplementary Fig. 5)

the RMSE is a factor three larger. The evaluated R-squared ($R^2$) shows, a model accuracy of 99%, i.e. demonstrates no underfitting or overfitting. Furthermore, the ML-based approach for the extraction of the deflection in comparison to conventional image analysis saves about 75% in time.

Figure 2b illustrates the analysed deflection obtained from the GRNN model along the thickness of the thin film. We depict for the deposited nanocrystalline films with 15 and 20 at% Ti a positive deflection. For 30 at% Ti we observe a negative deflection. This observed difference might relate to the observed microstructure disparity depicted in Fig. 1. Notably, the deflection profile changes significantly within the different $W_{1-x}Ti_x$ layer configurations and stays constant in the silicon oxide and Si layers (Fig. 2b).

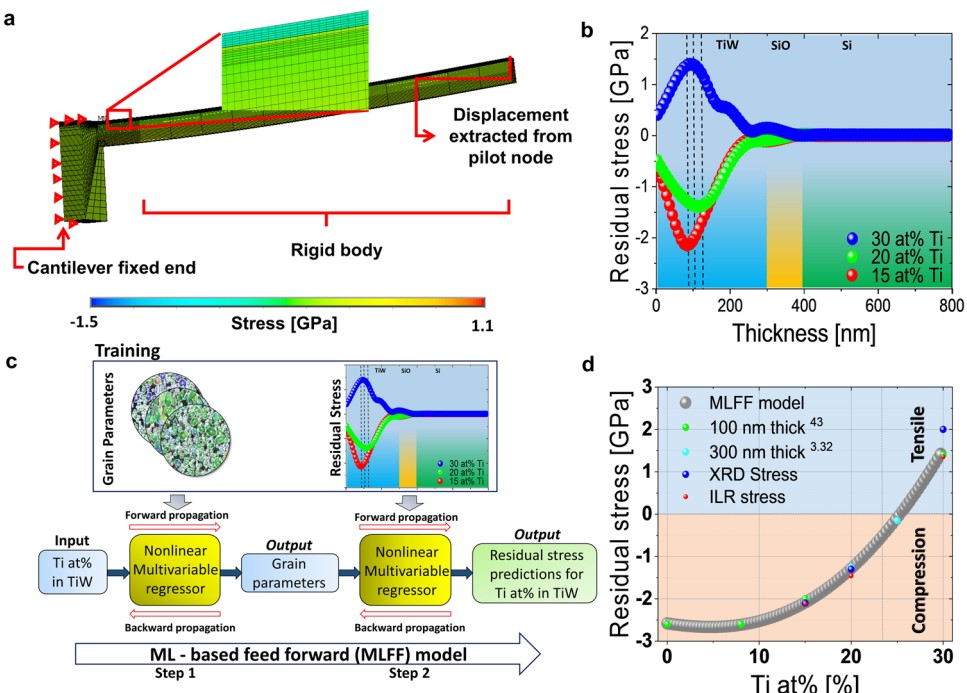

**Fig. 3 Residual stress evaluation from deflection data utilizing FEM and residual stress prediction. a** Numerically determined stress distribution projected on the micro cantilever beam (−1.5 to 1.1 GPa). A fine mesh with a mesh size of 10 nm, for the ILR-zone is applied in the FEM-modelling. **b** Numerically determined residual stress profile using the deflection evaluated by the GRNN-model for 15 at% (red), 20 at% (green) and 30 at% (blue) Ti, as a function of the layer thickness. A spline fit is used on the simulated stress profile data. **c** Schematic shows the workflow of the ML-based feed-forward model (MLFF) consisting of two connected multivariable regression (MR) architectures. Regime for compression and tension is highlighted. The connected models are used to provide an estimation about the minority element concentration where the crossing from compressive to tensile stress occurs. The input data for the first MR model is the Ti-concentration. Here, for the training, grain parameters are used. The grain parameters for the different Ti at% (x at% Ti, where x ∈ [0, 30]), which display the output of the first model, are used as an input for the second MR model. Experimentally evaluated residual stress data is used to train the second model. **d** Predicted, residual stresses between 0 and 30 at% Ti. We compare the prediction of the residual stresses by using the evaluated ILR-values along the thickness (red) for the training of the second MR-model. In addition, for comparison various literature[3,32,43] values for different at% Ti and the measured XRD values (blue) are illustrated. The different colours used for the literature values display measurements for different film thicknesses. The feed-forward model architecture predicts a crossing point, at about 26 at% Ti.

**Determination of the residual stress from the GRNN-predicted deflection using FEM.** The GRNN-predicted deflection data is used as an input for the finite element modelling (FEM). We extract the residual stress profile over the layer thickness with the resolution of 15 nm from the fully-functional nanocrystalline film. For this purpose, we develop a general model in the ANSYS mechanical parametric design language (MAPDL), see Fig. 3a, to overcome past evaluation limitations[12,13] e.g. in connection to orthotropic materials (see methods). Figure 3b, highlights the numerically evaluated residual stress distribution over the thickness of the different thin film stacks with 15 at% Ti, 20 at% Ti and 30 at% Ti. A spline fit is used to obtain the stress gradient and the residual stresses along the thickness. The distribution significantly changes over the $W_{1-x}Ti_x$ film thickness and differs for all three different sample configurations. We obtain a compressive stress distribution (positive deflection) for 15 and 20 at% Ti. A tensile stress distribution (negative deflection) is observed for 30 at% Ti. For 20 at % Ti, a maximum in the residual stress distribution is depicted at about 115 nm (measured from the surface). For 20 at% Ti the observed maximum is about 145 nm closer to the $W_{1-x}Ti_x$ / silicon oxide interface when compared to 15 or 30 at% Ti. For the 30 at % Ti sample we observe a shoulder at about 200 nm and a further increase of the residual stress at about 250 nm in the vicinity of the $W_{1-x}Ti_x$/silicon oxide interface. The sample with 15 at% Ti provides the highest absolute mean stress value (Table 2). For 30 and 20 at % Ti, the mean stress shows a similar value, however the values differ in sign. The residual stress distribution for the 20 at% is

broader compared to the 15 at% Ti film. By fitting the distribution with a Gaussian fit (see Supplementary Fig. 6) an FWHM of about 163 nm and about 134 nm is observed for 15 and 20 at%, respectively. Note, that the Si substrate is stress-free, i.e. shows zero for all three sample configurations (Fig. 3b).

To validate the obtained ILR results, we compare the evaluated ILR-mean value with results obtained from X-ray diffraction (XRD), see Table 2. The XRD, using the XRD-$\sin^2\psi$ method, provides an integral method to determine the residual stresses in $W_{1-x}Ti_x$ thin films globally. The mean residual stress determined from the ILR approach for 15 at% Ti and 20 at% Ti provides comparable values to the XRD result. However, for 30 at% Ti a deviation between the ILR mean residual stress and the XRD measurement with about 800 MPa is obtained.

To perceive an estimate about the crossing between the tensile and compressive stress with respect to the minority element

---

**Table 2 Comparison of residual stresses using the ILR and XRD method.**

| No | Sample | Residual Stress Results | |
|---|---|---|---|
| | | **XRD for $W_{1-x}Ti_x$ stack** | **Mean value ILR for $W_{1-x}Ti_x$** |
| 1 | 30 at% Ti | 2032 MPa ± 203 MPa | 1279 MPa ± 289 MPa |
| 2 | 20 at% Ti | −1328 MPa ± 103 MPa | −1211 MPa ± 146 MPa |
| 3 | 15 at% Ti | −2110 MPa ± 153 MPa | −2047 MPa ± 389 MPa |

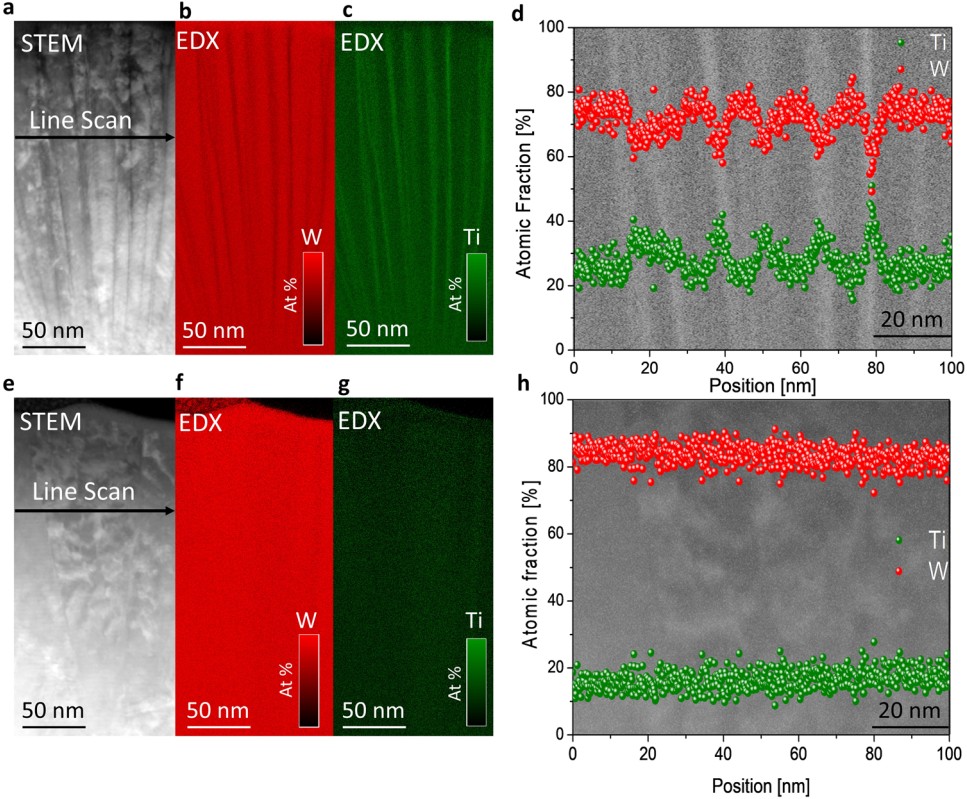

**Fig. 4 ADF STEM and STEM-EDX analysis of the 30 at% and 20 at% Ti nanocrystalline W$_{1-x}$ Ti$_x$ thin films. a** Cross-sectional ADF STEM image for 30 at% Ti. **b** W element map (red) for 30 at% Ti. **c** Ti element map (green) for 20 at% Ti. **d** Projected line scan for W (red) and Ti (green) on the ADF STEM image for 30 at% Ti. **e** Cross-sectional ADF STEM image for 20 at% Ti. **f** W element (red) map for 20 at% Ti. **g** Ti element map (green) for 20 at% Ti. **h** Projected line scan for W (red) and Ti (green) on the ADF STEM image for 20 at% Ti.

concentration, we develop a ML-based feed-forward model (MLFF) consisting of two connected multivariable regression (MR) architectures. The first model is trained using the microstructure data evaluated from the FESEM-EBSD images see Fig. 3c, such as minor axis of the fitted ellipse, major axis of the fitted ellipse, aspect ratio of the fitted ellipse, grain density and area of the fitted ellipse, number of grain boundaries along the film thickness. The input to the first MR model is $x$ at% Ti, where $x \in [0, 30]$. This first MR model enables us to predict the grain parameters for $x$ at% Ti. Furthermore, the generated output from the first model is used as an input for the second MR model within the ML-based architecture. The second MR model is trained using the locally evaluated residual stresses along the nanocrystalline film for 15, 20 and 30 at% obtained by the ILR-method. Finally, the second model predicts the residual stress for different minority element concentrations. Figure 3d illustrates the predicted non-linear correlation between the residual stress and the Ti at% indicating a crossing point between compressive and tensile stress at about 26 at% Ti. We compare the data-base driven prediction of the residual stress with literature values[3,32,43]. The model provides a RMSE of 67 MPa and $R^2$ of 99%.

**Elemental analysis of the nanocrystalline W$_{1-x}$ Ti$_x$ thin films.** In addition to the microstructure analysis we perform elemental analysis, for the cross-sectional (x-z plane) data, conducting scanning transmission electron microscopy-energy dispersive X-ray spectroscopy (STEM-EDX), to understand the W-Ti distribution within the nanocrystalline alloy. In Fig. 4a, we show the annular dark-field scanning transmission electron microscope (ADF STEM) image from the cross-section for the 30 at% Ti

sample. The observed columnar structure indicates a mean width of approx. 20 nm (in x-direction), similar to the mean value evaluated from the FESEM-TKD data shown in Table 1.

In Fig. 4b, c, we obtain the elemental distribution of W (red) and Ti (green) by using STEM-EDX for the 30 at% Ti sample, respectively. Figure 4d provides further information about the element variation along a defined line scan shown in Fig. 4a. Significant fluctuations in the element distribution along the x-direction are observed. The mean Ti atomic concentration is 28.2% ± 4.2% for the nanocrystalline thin film. The maximum and minimum of Ti concentration is about 50.8% and 12.0%, respectively. The mean for all the maxima and the minima in the line scans of the Ti is 41.8% and 18.6%, respectively. The observed fluctuations of the local Ti content increase by almost 20 at% in the vicinity of the grain boundaries throughout the measured data.

In Fig. 4e, we show the cross-sectional ADF STEM image of 20 at% Ti. The elemental distribution obtained from STEM-EDX is shown for W (red) and Ti (green) in Fig. 4f, g, respectively. The sample with the nominal 20 at% concentration exhibits a larger grain extension (x-direction) compared to the 30 at% Ti sample. The mean width of the grain is about 50 nm (x-direction). This is in the line with the mean value evaluated from the FESEM-TKD data shown in Table 1. In Fig. 4h, the corresponding element distribution along the defined scan in Fig. 4e is shown. No significant fluctuations along the x-direction as shown for 30 at% Ti are observed. It provides for the atomic concentration of Ti a mean value of about 18% over the defined line scan width. In addition, orientation maps, see Supplementary Fig. 7, are collected as a useful complement to the ADF STEM contrast images presented in Fig. 4a, e. Hence, we gain further information about the observed fine structure and confirm the polycrystalline

structure of the 20 and 30 at% Ti thin film. Different grey values as seen in the TEM contrast images, see Fig. 4a, e, indicate the separation of the grain boundaries with differently oriented crystal orientations.

For all investigated Ti samples, the observed mean value of the concentration corresponds well with the targeted Ti-concentration. The grey value images (Fig. 4a, e) illustrate for the 30 at% Ti as well as for the 20 at% Ti alloy a V-shaped morphology i.e. grains are small on the bottom and larger at the top of the film (zone T)[26,30]. Commonly, this can be understood, that the grain boundaries become less mobile for larger grains as the film gets thicker during growth[1,46].

**Atomistic modelling of segregation behaviour**. In order to address the segregation state observed in Fig. 4, we carried out density functional theory (DFT) simulations of the Ti segregation the $\Sigma3[110](\bar{1}11)$ grain boundary (GB) that is representative for high angle and high energy GBs[47–49]. Although the real coating contains many different grain boundary orientations which may exhibit relative differences in GB segregation state, we expect that the general segregation trend reported below remains the same because detailed analysis of segregation at various microstructure defects shows only small changes in the distribution of segregation energies[24,50].

Figure 5a shows the DFT results where different Ti configurations with increasing GB-Ti content are considered. Due to configurational differences (see Fig. 5a i-vi), a considerable spread in segregation energies up to about 1 eV can be observed at a given GB concentration. Additionally, a clear trend towards higher segregation energies can be recognized at higher GB-Ti contents. Importantly, all segregation energies are positive which implies that the GB is not energetically favourable for the solute and vice versa. Therefore, as long as segregation is considered in the dilute limit, no enrichment of Ti at grain boundaries is expected from DFT simulations.

The situation changes once the Ti concentration in bulk increases. To understand the reason for this behaviour we focus on the formation energy of bcc $W_{1-x}Ti_x$ in bulk and at the GB. Note that such a diagram has been sketched before in the context of nanocrystalline stability[34] but was not predicted based on ab-initio simulations. As can be seen in Fig. 5b, the bulk formation energy taken from ref. [37] is negative up to concentrations of about 50 at% Ti with a clear minimum at 12.5 at%. This implies that Ti prefers to be coordinated by W atoms in this concentration range. The formation energy of the grain boundary shows quite a different trend. In general, the offset between the curves is equal to the grain boundary energy per GB atom. The curve shows a monotonic decrease of roughly parabolic shape. The segregation energy in the concentrated solution is given by the difference in the first derivative of the two formation energy curves (see Supplementary Note 3 and Supplementary Fig. 8). The result is shown in Fig. 5c where the segregation energy for different bulk concentrations is shown as function of the Ti-GB concentration. For a low Ti-bulk concentration, the curve reproduces largely the segregation energies shown already in Fig. 5a. For higher Ti-bulk concentrations, the curve shifts towards lower values indicative of the attractive binding of Ti to the GB. With the use of a thermodynamic model (see Supplementary Note 3), the GB concentration is calculated for different temperatures. The resulting GB excess, i.e. the increases of Ti atoms at the GB compared to the bulk, is shown in Fig. 5d. Due to the particular dependency of the segregation energies with bulk concentration, a non-linear increase in GB excess is observed between 0 and 30 at% Ti where the excess shows a maximum. Notably in this context is also the intensified change of the slope above about 20 at% Ti.

For higher Ti-bulk concentrations, the excess decreases almost linearly to zero. To highlight the importance of this results we also show in Fig. 5d the simplified calculation were the segregation energy is fixed to −0.5 eV irrespective of bulk concentration as was assumed in ref. [30]. In this case the GB excess would lead to a linear decay over the whole concentration range (see Fig. 5d). GB excess in the region between 0 and 30 at% Ti is, thus, strongly governed by concentration dependent interactions between Ti and W atoms in the bulk and at the GB.

## Discussion

As stated in ref. [26,51] the emerging stress in the nanocrystalline thin films (thickness <1 μm) depends strongly on various factors relating to the microstructure like the grain geometry, the grain size, crystallographic orientations of the grains, surface mobility and the grain growth conditions (coalescence). The surface analysis as shown in Fig. 1b suggests a specific crystallographic orientation along [110] for 15 and 20 at% Ti. Although the crystallographic orientation on the surface is similar for 15 and 20 at% Ti, the grain density for 20 at% Ti is about 19% smaller than for the 15 at % Ti sample (Fig. 1b, Table 1). We argue, also in agreement with[25,51,52], that the observed grain size difference between 15 and 20 at% Ti might be related to the higher compressive stress. The employment of the GRNN model to extract efficiently the deflection from the SEM image data, which is further used to extract the local residual stress along the film applying FEM, provides high accuracy and efficiency. According to Fig. 3b the maximum of the observed stress is 20% higher for the 15 at% Ti sample. For 15 and 20 at% Ti the obtained ILR mean stress, as shown in Table 1, corresponds well with the XRD results. The observed difference in the mean stress for 30 at% might be linked to the comparatively inhomogeneous lamella distribution illustrated in Fig. 1a. The ILR-method, using the micro-beam setup, records locally with high resolution the stress distribution, whereas XRD on the other hand displays an integral method collecting an averaged stress value over a defined surface area, here with 1 x 1cm². The evaluated stress profile provides not only information with respect to the tensile or compressive stress states but also show additional important information about the stress maximum location with respect to the silicon oxide interface and stress profile shape. The maximum in the stress distribution in Fig. 3b for the 20 and 15 at% Ti is located at 115 nm and 95 nm (measured from the thin film layer surface), respectively. That is, for the 20 at% Ti the stress maximum (compressive) is closer to the silicon oxide surface. We argue in accordance with[32] that the detailed knowledge about the stress distribution within the thin film is important in the context to the adhesion properties of the thin film.

The elemental analysis in Fig. 4 demonstrates a rather heterogeneous W-Ti distribution for 30 at% Ti where Ti-segregated zones clearly emerge along the grain boundaries (Fig. 4d). For 20 at% Ti, rather a homogenous distribution of W-Ti is observed in the microstructure according to the STEM-EDX data depicted in Fig. 4h. Both, the 20 at% Ti as well as the 30 at% Ti thin film show in contrast to 15 at% Ti, a columnar grain structure in the x-z plane (cross-section), see also Fig. 1e and Table 1. The microstructure analysis shows that the grain density obtained from the cross-sectional and surface data is more than a factor two larger for the thin film with 30 at% Ti than for the one with 20 at% Ti (Table 1). According to[3] the stress state is altered by changing the grain boundaries and/or the number of boundaries in the film.

To understand the microstructure and elemental composition data in the context to the observed residual stress we further apply DFT calculations. As shown in ref. [37], alloying Ti into W leads to a nonlinear change in lattice parameter. While for the

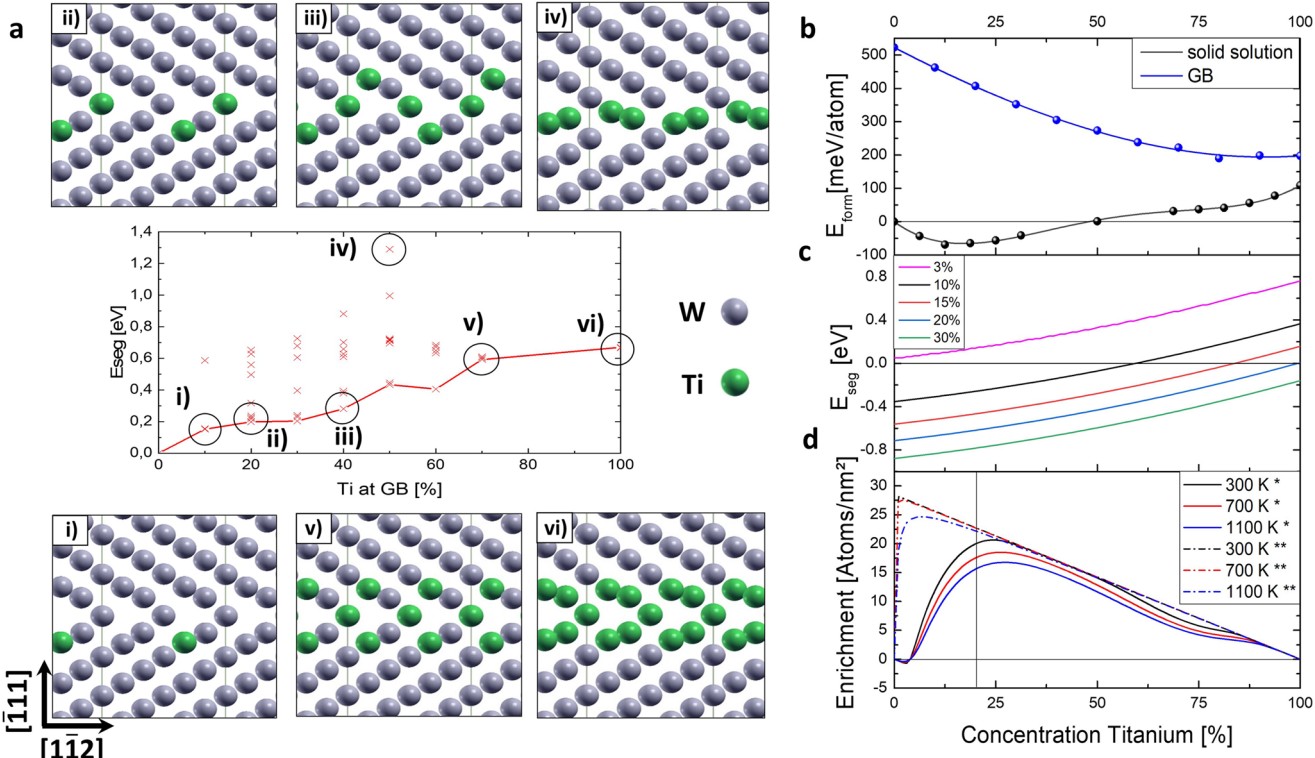

**Fig. 5 Atomistic modelling of the Ti segregation and enrichment as well as the formation energy of W$_{1-x}$Ti$_x$ in bulk and at the grain boundary. a** GB segregation at the dilute limit with different Ti concentrations from (i) to (iv), as well as the corresponding illustrated Ti-atoms distribution (green) at the GB. **b** Formation energy of the GB- and Bulk-WTi. The Titanium concentration on the x-axis refers to GB and bulk Titanium concentration. **c** Segregation at fixed bulk Ti concentration. The Titanium concentration on the x-axis refers to GB Titanium concentration. **d** Ti enrichment of the GB. The * and ** in the legend represent averaged E$_{seg}$ and E$_{seg}$ = −0.5 eV, respectively. The Titanium concentration on the x-axis refers to the bulk Titanium concentration.

concentration range, up to about 20 at% Ti, the lattice parameter is remarkably insensitive to the Ti concentration indicative of the fact that the Ti atom is of about the same size of W in this concentration regime. Beyond this regime the lattice parameter increases steeply showing that the size of Ti is larger than W in the bcc lattice. This assumption was also reported in previous works[30]. Therefore, we can assume that in the low concentration range segregation is not associated with a strong volume change. In contrast, at bulk concentrations higher than 20% segregation is associated with a reduction in volume and, consequently, tensile stresses. For the 30 at% Ti alloy, the applied ILR method indeed demonstrates tensile stresses along the W-Ti film (Fig. 3b). Whereas, changing the Ti concentration to 20 at% Ti leads to compressive stress (Fig. 3b) within the thin film. According to the ML-based feed-forward prediction model which is constructed of two consecutive regression architectures and fed by microstructure and residual stress data, respectively a crossing between the compressive and tensile stress is observed at about 26 at% Ti, see Fig. 3d. Our simulation approach explains the high segregation levels observed in experiment at 30 at% Ti. Furthermore, we can understand why Ti enrichment is strongly reduced at smaller concentrations. While we cannot perfectly reproduce the low segregation levels of the experiment at 20 and 15 at% Ti, our DFT approach reveals the right trend and the strong sensitivity of GB excess with the GB concentrations. The reason why GB excess is higher in DFT compared to experiment could be (i) the omission of vibrational contributions in the calculation of formation energies, (ii) the chosen mean field approximation, (iii) noise in the measurement and (iv) the fact that experiment is not in equilibrium. Reason iv is the most likely to explain the discrepancy since another experiment carried out also for 20 at

% Ti[30] did observe substantial increase in GB excess in agreement with our DFT result.

Note that there have been previous attempts to explain segregation behaviour in W-Ti based on Monte Carlo simulations[30]. Our results cast doubts onto the assumptions used to parametrize the bond energies of this previous approach. The segregation energies were assumed to be −0.5 eV (with the present sign convention) even in the dilute limit because the simplified Miedema model was used in[30]. Our results show that segregation energies are strongly dependent on the bulk content and are positive in the dilute limit. As highlighted here, only at larger Ti concentrations GB segregation energies become negative. Furthermore, formation energies in[30] were taken from a previous CALPHAD assessment[53], where the formation energy is positive at all concentrations. Also this assumption has been proven wrong by recent ab-initio simulations[37,54]. Therefore, we argue that our model is based on correct thermodynamic data and reveals the true origin of segregation in the W-Ti system.

## Conclusion

In summary, the unique framework linking experimental, and modelling methods as well as machine learning assisted analysis provides insights into how the minority element concentration within nanocrystalline thin films impacts the residual stresses generated after the deposition process. Ti enrichment can be strongly reduced at smaller concentrations and significantly affects the stress stored in the nanocrystalline thin film. Thus, the consideration of the concentration dependent interaction between the minority element and host element in the bulk and at the grain boundary is essential. The correlation of local residual stress

behavior with microstructure as well as considerations on atomistic level is needed to improve nanocrystalline thin films. Employment of machine learning for the local residual stress evaluation with ILR, a method which is generalizable across materials, as well as for predicting the stress over various minority element concentration displays expanded possibilities with respect to efficiency and accuracy. The presented unique approach renders possible development of nanocrystalline thin films with enhanced functional properties for future engineering applications.

## Methods

**Samples**. There are 3 samples of $W_{1-x}Ti_x$ at hand for comparison of stress profiles as a function of thickness and respective morphological changes and differences. We fabricate 30, 20 and 15 at% Ti concentration using different deposition conditions (Physical vapour deposition) in order to achieve these different concentrations of Ti in $W_{1-x}Ti_x$. All the samples are deposited on Si <100> wafers, non-stoichiometric silicon oxide (~100 nm thick) and TiW (~300 nm thick).

**Sample preparation for ILR**. A Hitachi-E3500 Cross-section polisher (Hitachi, Tokyo, Japan) on the area of 80–100 μm is used where low energy Ar ions polishes off the front edge of the sample to eliminate any deformed material remaining from breaking the wafer. Advantage of this procedure compared to FIB milling are less ion damage of the sample and a shorter preparation time. In the second step, the sample is loaded in AURIGA 40 Crossbeam Workstation to make the final beam geometry of the beam by milling from the top and the bottom. The final cantilever is of approximately 100 μm long and cross-section of $5 \times 4 \, \mu m^2$.

**ILR method—experimental**. Figure 2a, shows the SEM-SE image of the milled micro cantilever exemplarily for 20 at% Ti. First, a free-standing structure is pre-milled with an Hitachi argon ion-slicer and then further prepared with a SEM/FIB setup[12,13,19]. For the ion layer removal (ILR) method a Zeiss Auriga 40 Crossbeam Workstation is used. Focused ion beam of Ga ions is used to mill away the layers in ILR zone. Once the free-standing structure is milled, a micro-cantilever is fabricated using a focused ion beam (FIB) to make one end loose from the bulk. Subsequently the initial deflection of the cantilever is measured and shall not exceed a certain value (about 2 μm) to avoid plastic deformation. This observed initial deflection can be associated with the global residual stress stored in the micro-cantilever. To gain information about the locally stored residual stress distribution over the thickness of the micro-cantilever beam, we remove with an ion beam within a defined area (size: $5 \times 4 \, \mu m^2$), layer by layer. The ILR-zone is about 95 μm away from the loose end of the cantilever beam. The cutting depth interval is about 15 nm. The change in thickness leads to a change in the deflection of the cantilever beam. We apply a self-developed fully automated cutting routine. This routine works as follows: First a layer with the desired depth is removed from the ILR area, in our case 15 nm per step (FIB specifications: current ~ 50 pA, Voltage ~ 30 kV). After removing each layer, images are taken from four different positions of the micro-beam to analyse the data accurately. Two different detectors namely, Secondary electron (SE) and the Back Scattered Electron (BSE) detector are used. The same steps are repeated until we reach the substrate.

**ILR method—FEM Simulations**. We carry out finite element simulations using ANSYS 17.2 MAPDL. In combination with Python 2.7, an automatized optimization routine is created to simulate the complete ILR experiment for all the steps. The beam geometry is divided in two regions (see Fig. 2a). The first region is a region where the bending changes during the process of removing layers in this region (the ILR region). The second one is the region where no material is removed and therefore the curvature stays constant during the experiment and which solely acts as an indicator amplifying the deflection stemming from the curvature change in the ILR region. The developed numerical approach is not limited to a certain type of material or material system. We use a general parameterized model[55–58], which can be adapted with minimal effort to any material system. Therefore, an iterative bottom up approach for the FEM with an optimization routine was used (see Supplementary Note 4). Once the geometry is generated, mesh convergence[55] is carried out to find a suitable element size for the model (Fig. 3a). The first region has to be meshed with a sufficiently fine grid, whereas the second region can be modelled in a coarse way and by defining it as a rigid body. For the meshing of the interface between the two regions, we use contact elements—Surface to Surface[55,56]. This approach provides an even and well-connected interface with a reduced number of equations (see Supplementary Note 4 and Supplementary Fig. 9). An optimization subroutine is implemented where trial initial stress values are used as an input for the simulation. At first a layer is deposited in the ILR zone during the first optimization iteration. Only one optimizing parameter for each layer is necessary which makes the optimization very stable as well as time efficient when compared to previous approaches[13,44]. For each deposited layer in the ILR-zone a Regula-Falsi method[59], commonly known as bisectional method / false position method, is applied iteratively until the deflection value is determined up to prescribed accuracy. Once the deflection for the individual layer i between

simulation and experiment matches, the optimization for the subsequent layer $i + 1$ in the ILR zone follows. The simulation keeps running till the surface is reached.

**EBSD-analysis**. EBSD is carried out using a field emission scanning electron microscope Zeiss 450 Gemini-SEM in combination with an Oxford detector with $1244 \times 1024$ pixels. We perform the EBSD measurements in reflection (FESEM-EBSD) as well as in transmission mode (FESEM-TKD). For the transmission measurement, a lamella was fabricated using the Zeiss Auriga 40 Crossbeam Workstation. Current and voltage settings for the resulting images are 20 nA (up to 500 pA) and 20 kV (up to 5 kV), respectively. In reflective mode we achieve 10 nm pixel size. In transmission mode a pixel size of 0.5 nm is achieved. The sample holder is placed at 70° angle from the detector. Hough resolution is 96 and the magnification is x20k. We use the software AztecCrystal by Oxford instruments, to retrieve precise boundary statistics. In particular, we evaluate the grain shape for the surface and cross-section data by fitting the grains with ellipsoids (see Supplementary Fig. 3 and Supplementary Fig. 4). The total number of grains is used to evaluate the grain density. All depicted values are summarized in Table 1.

To retrieve information about the grain size from the surface EBSD data, we quantify the largest extension of the grains i.e. the maximum measure of the object size between two defined parallel lines.

**TEM—analysis**. STEM imaging and EDX analysis is performed using a Thermofischer Titan Themis TEM. The microscope is running in a scanning mode at an accelerating voltage of 300 kV. The beam current is set to 0.1 nA for imaging and 1nA for the EDX analysis. The images are recorded using an annular dark-field detector (ADF). The STEM-EDX data is recorded using a ChemiSTEM technology utilizing four SUPER-X windowless SDD detectors and subsequently evaluated in a Velox software. The selected area electron diffraction (SAED) patterns were taken using 200 nm SAED aperture at a camera length of 600 mm. In total five SAEDs were taken from different regions from each sample and integrated to obtain orientation maps (diffraction patterns). Periodic line scans are carried out starting from the substrate with an interval of 30 pixels. The pixel size is about 0.167 nm. In total, 55 scans are performed along the z axis in the cross-section.

**Image analysis**. We perform image processing of the gained FESEM data using Python libraries including Numpy, Scipy, and Scikit-image. We used Avizo® (version 2019.1) for rendering (see Supplementary Note 3).

**Regression neural network for the deflection characterization**. A regression neural network is designed and used in order to obtain the deflection data using Python 3.6. Images are of size $2048 \times 1536$ pixels. Images are pre-processed using Histogram of gradients features extraction. These features are then used for principal component analysis and these pre-processed features are then given to the network as input. The activation function of the network is logistics. The model consists of three hidden layers and four outputs. 350 to 370 neurons, 250 to 270 neurons and 100 to 170 neurons are used for the 1st, 2nd and 3rd hidden layer. The activation function of the model is logistics and the solver used is the limited-memory Broyden–Fletcher–Goldfarb–Shanno algorithm (LBFGS)[45].

**Non-linear multivariable regression neural network for the stress prediction**. A non-linear multivariable regression neural network is designed in order to obtain the stresses using python 3.6. Grain parameters of surface and cross section are separated for the training of the model. This grain parameters include the number of grain boundaries in the cross section in correlation to the ILR steps. First, the model predicts the grain parameters of x at% Ti, where x is between 0 and 30. Those parameters in combination with the ILR stress values are fed to the same model and we get the residual stresses for x at% Ti sample.

**DFT—analysis**. For DFT calculations, we used the Vienna ab initio simulation package with periodic boundary conditions and the projector augmented wave method (PAW)[60–63]. For the grain boundary calculations, we employed the same input parameters as for the bulk calculations, namely k-point densities corresponding to a k-point mesh of 20 x 20 x 20 for the conventional 2-atom cubic bcc W cell, an energy cut-off of 400 eV, as well as 12 and 10 valence atoms for W and Ti, respectively. Detailed computational details are given in ref. [37] and the methodology for segregation is explained in the Supplementary Note 3.

## Data availability

The data sets generated/analysed during the current study are available from the corresponding author on reasonable request.

## Code availability

The codes developed during the current study are available from the corresponding author on reasonable request.

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

## Acknowledgements

The authors gratefully acknowledge the financial support under the scope of the COMET program within the K2 Center "Integrated Computational Material, Process and Product Engineering (IC-MPPE)" (P. No.: 886385, P2.12 and P1.12). This program is supported by the Austrian Federal Ministries for Climate Action, Environment, Energy, Mobility, Innovation and Technology (BMK) and for Labour and Economy (BMAW), represented by the Austrian Research Promotion Agency (FFG), and the federal states of Styria, Upper Austria and Tyrol. B. Sartory is thanked for his help on ILR experiments and EBSD/TKD measurements. L. Romaner and D. Scheiber acknowledge funding by the Austrian Science Fund (FWF), P. No.: 34179. This work has been carried out within the framework of the EUROfusion Consortium, funded by the European Union via the Euratom Research and Training Programme (Grant Agreement No 101052200—EUROfusion).

## Author contributions

R.B. and R.J.S. planned the ILR experiments. R.J.S. performed the ILR and EBSD analysis under supervision of R.B.. R.B., R.J.S. and F.F.C. planned and performed the image analysis work. R.J.S. developed the FEM simulations under guidance of R.H.. M.R. provided the samples. P.P. set up the ML-model under supervision of R.B.. J. Z. and J.K. conducted the XRD and STEM-EDX measurements. R. Bo., D.S., and L.R. planned and conducted the DFT simulations. R.J.S. and R.B. wrote the manuscript with input about the DFT section from D.S., R. Bo. and L.R. All authors discussed the results and commented on the paper.

## Competing interests

The authors declare no competing interests.
