## [Peer Review File · Communications Materials]

28th Jun 22

Dear Dr Brunner,

Thank you for submitting your manuscript, "Machine-learning assisted local stress evaluation and atomistic modelling to inquire Ti-enrichment in W-Ti nanocrystalline films", to Communications Materials. It has now been seen by 3 referees, whose comments are appended below. You will see that while they find your work of some interest, they have raised various substantial concerns. We cannot accept the manuscript for publication and it is unclear to us whether you can address their comments. However, we may be willing to consider a revised version that addresses these serious concerns.

You will see that the referees collectively raise various issues, including the contribution from the machine learning part being small and possibly lacking in accuracy, limitations of the DFT calculations, and a large amount of additional experimental data and analysis being needed. Indeed, the extent of changes needed possibly goes beyond what could be expected from a revision, and you may wish to seek publication elsewhere rather than submitting a revision to us.

However, should new data and analysis allow you to address these criticisms, we would be willing to look at a substantially revised manuscript. Please bear in mind that we will be reluctant to approach the referees again in the absence of major revisions. If the revision process takes significantly longer than three months, we will be happy to reconsider your paper at a later date, as long as nothing similar has been accepted for publication at Communications Materials or published elsewhere in the meantime.

When submitting your revised manuscript, please include the following:

-A response letter with a point-by-point reply to each of the referee comments and a description of changes made. Please include the complete referee report in the response letter. Please note that the response letter must be separate to the cover letter to the editors.

-A marked-up version of the manuscript with all changes to the text in a different colored font. Please do not include tracked changes or comments. Please select the file type 'Revised Manuscript - Marked Up' when uploading the manuscript file to our online system.

-A clean version of the manuscript. Please select the file type 'Article File'.

-An updated <https://www.nature.com/documents/nr-editorial-policy-checklist.zip> Editorial Policy checklist, uploaded as a 'Related Manuscript File' type. This checklist is to ensure your paper complies with all relevant editorial policies. If needed, please revise your manuscript in response to these points. Please note that this form is a dynamic 'smart pdf' and must therefore be downloaded and completed in Adobe Reader. Clicking this

link will download a zip file containing the pdf.

Please use the following link to submit your revised manuscript files:

[link redacted]

We understand that due to the current global situation, the time required for revision may be longer than usual. We would appreciate it if you could keep us informed about an estimated timescale for resubmission, to facilitate our planning. Of course, if you are unable to estimate, we are happy to accommodate necessary extensions nevertheless.

Please do not hesitate to contact me if you have any questions or would like to discuss the required revisions further. Thank you for the opportunity to review your work.

Best regards,

Xiaoyan Li, PhD
Editorial Board Member
Communications Materials
orcid.org/0000-0002-2953-9267

Reviewers' comments:

Reviewer #1 (Remarks to the Author):

This manuscript deal with the build up of residual stresses in thin films using experimental and modeling data analyzed by so called "machine learning". The topical area is interesting, but I find the manuscript approximate and that the authors are failing to meet the target objective by a large margin. Below are some comments for the authors to consider.

1. Reading the introduction, I am a bit surprised the authors d not refer more explicitly to the body of work by Eric Chason on residual stresses. His group developed mechanistic based model to predict residual stresses based on curvature measure (improvement of the Stoney's equation) using various conserdations on the origin of the residual stresses in the film.

2. Also still in the introduction, the topic of solute segregation to stabilize thin film is well know see work from Boyce and co-workers or Schuh and coworkers. In either case, there are many example of experimental and computational studies, some recent (see Acta Mater 2021/2022?) some older (back in the mid 2010s). The explanation in the second paragraph is at best hand-waving.

3. I find the use of vocabulary "incorporating machine learning assisted" really vague. It

would be like saying we solve a data-driven problem using computer programming. I would strongly suggest that the author revise their manuscript and be more precise and to the point than saying "machine-learning" as it does not convey ANY useful information to the reader AND gives the impression that the authors are using it as a black box tool not necessarily knowing/understanding what they get out of it.

4. I find the description the ML model confusing. The authors mix details about the experimental set up (lines 79-88), some information on so called "trained neural net" (lines 88-102) and then back to some experimental consideration (lines 103-105). I would encourage the authors to rethink this section. This goes back to my comment "3" about the lack of precision in stating "machine learning". The authors seem to be interested in a supervised learning problem with labels. I would clearly state: what is the input, what is the output, what is the intended task at play, and what are the different components used to perform that task. It is unclear reading the subsection how PCA, HOG, SEM and FEM data are all linked together...As for the original draft of the paper, the write up does not give confidence that the authors know how to properly use this information.

5. On the FEM model. What not used the closed form "expert model" from the work from Eric Chason. In other words, the authors are basically looking at a multi-fidelity approach (combining experimental data [high-fidelity] with low fidelity data [FEM]) to infer correlations. See comment 1. If anisotropy and other factors are required from the FEM that is not available otherwise then the authors should clearly contrast their approach with known models.

6. Please check spelling (see for example line 150 "previously"). There are more throughout the manuscript.

7. The microstructure analysis could be interesting, but without any mention of the PVD conditions that lead to such texture and grain morphology, it is not that useful. It is well known that depending on the background gas pressure, sputtering power of the targets used, working distance we could have (i) very different microstructures and (ii) very different residual stress states. I am assuming here that the authors work at fixed conditions? Are their changes in the sputtering power for the different at. % of Ti?

8. The observations of GB segregation is consistent with computational model (for instance phase-field, see work by Abdeljawad and co-workers). I would recommend that the authors make a note of that and link their observation with known GB segregation mechanisms. Especially if they observed graded compositions.

9. Can you please add labels of the GB studied in Fig. 4? Also it is unclear if only the W sublattice is shown or else.

10. On the GB segregation. Could the authors comment on the relevance of their simulation results as compared to experimental results especially since deposited GB are most likely not in their ground state (as it is the case in the simulation) and therefore segregation energetic landscape would differ substantially for defected GB as compared to ground-state one (see for instance work by Schuh and coworkers).

11. Overall the whole analysis of using "machine-learning" to correlate residual stress with composition fall way short of the stated objective. The paper only has a very short section on the "ML" approach" and the rest of the analysis is rather a classic experimental/modeling analysis of the collected data.

Reviewer #2 (Remarks to the Author):

The manuscript presented an interesting study on the evolution of residual stress in W-Ti films. Ti solutes exist in supersaturated solid solution for 10 and 20%Ti, whereas Ti solutes segregate to grain boundaries. In parallel the residual stress in the films change signs and varies drastically. Extensive EBSD studies have been performed. While these experimental studies and modeling add something new to the literature, the following concerns must be addressed.

1. The authors contended that nanocrystalline metals have poor thermal stability, while this is probably true for many cases, such a phenomenon does not constitute a good motivation for the study of residual stress in W-Ti films. For instance, there is no attempt made to investigate the thermal stability of their films. Hence the introduction section should be rewritten to fit in the scope of the manuscript.
2. Given the fact that the sign of residual stress switched from compression to tension between 20 and 30% of Ti, it is unnecessary to investigate a film with intermediate chemistry (25%) to probe the evolution of residual stress.
3. Although cantilever beams have been used to quantify residual stress, other methods, such as x-ray diffraction and curvature measurements on conventional (flat) Si substrates have been routinely used to measure residual stress, and one of these methods should be used in parallel to validate their measurement.
4. It is unusual to put discussion and conclusions together. The conclusions must be put into a separate section and serve as a concise summary of major findings.
5. There is no detail on film deposition: deposition method, vacuum of the instrument, substrate temperature etc.
6. STEM has been shown, but TEM evidence is lacking. The diffraction of TEM micrographs should also be provided and indexed.
7. Abstract: "We inquire why the experimental observed Ti enrichment can be strongly reduced at smaller concentrations and discuss how it significantly effects the stress stored in the nanocrystalline thin film." Effect is a noun, not a verb.

Reviewer #3 (Remarks to the Author):

In the manuscript "Machine-learning assisted local stress evaluation and atomistic modelling to inquire Ti-enrichment in W-Ti nanocrystalline films", the authors presented a framework containing experimental, numerical and ML methods to study the W-Ti alloy thin films. The residual stress, microstructure, elemental distribution and segregation behavior are clearly

studied. The paper is novel and well written. I recommend publication with some revisions.

1. The results contain too much detail descriptions of simulation methods, which can be moved into the Methods.
2. The lamella density and grain density need to be transformed to size, for better comparison with other studies. Besides, what are the lamella structures at microscale? Their density is much higher than that of grain density, and also does not match with the columnar density observed in Figure 3 (20-30 nm for 30% sample corresponding to 1000-2500 μm^{-2}).
3. Machine learning has become a powerful tool to study the structures and properties of materials. However, it seems that ML is not irreplaceable in this paper. ML is employed to extract the micro-cantilever deflection, which actually can be obtained in many ways. Labeling training images can also be time-consuming, reducing the efficiency of ML. I suggest the authors to directly comparing the accuracy and efficiency of ML framework (including data preparation and model training) with human-annotated methods.
4. To the best of our knowledge, training a ML model with images often requires a large training dataset. How to confirm that the ML model is well trained without overfitting or underfitting? The predictive error (root mean square error and R^2) as a function of training data size should be supplemented.
5. A simple $\Sigma 3$ grain boundary was selected for DFT calculations. It is understandably owing to the intrinsic limitations in DFT. But it is well known that the property of grain boundaries can be significantly influenced by their structures like misorientation angle. The authors could calculate segregation energies of some other grain boundaries with low Σ values. On the other hand, a rough statistic for the misorientation angle distributions in the samples can be helpful to determine the GBs to be calculated.
6. In Figure 3, a thick layer of segregation occurred at the boundary (few nanometers), but in the DFT simulation only 2-3 atomic layers is considered.

Dear Referees,

Please see our detailed response point by point for each referee below.

Reviewer #1 (Remarks to the Author):

This manuscript deal with the build up of residual stresses in thin films using experimental and modeling data analyzed by so called "machine learning". The topical area is interesting, but I find the manuscript approximate and that the authors are failing to meet the target objective by a large margin. Below are some comments for the authors to consider.

We thank Reviewer 1 for her/his questions and remarks and finding the topic interesting. In the following, we will address each question point by point.

Q1. Reading the introduction, I am a bit surprised the authors d not refer more explicitly to the body of work by Eric Chason on residual stresses. His group developed mechanistic based model to predict residual stresses based on curvature measure (improvement of the Stoney's equation) using various conderations on the origin of the residual stresses in the film.

Thank you for the comment. Although we referred previously to the work of Chason, we addressed the comment and think that the introduction improved significantly by adding additional text and references in this context. In addition, we referenced E. Chason's nice work as suggested by the referee in more detail. In particular, try to point out more clearly the main differences between Chason's approach and our approach. Our approach provides a possibility to locally witness the stress properties within the thin film after the deposition is concluded. The development is driven to understand the stress distribution in the fully functional thin film enabling to compare after the deposition differently cured films for life cycle analysis. In particular, we are able to gain information about the position of the maximum stress in relation to the interfaces, the overall stress distribution, local stress along the thin film as well as the local assignment of tensile and compressive stress after the deposition has been concluded.

Q2. Also still in the introduction, the topic of solute segregation to stabilize thin film is well know see work from Boyce and co-workers or Schuh and coworkers. In either case, there are many example of experimental and computational studies, some recent (see Acta Mater 2021/2022?) some older (back in the mid 2010s). The explanation in the second paragraph is at best hand-waving.

We agree with the reviewer that the topic of nanocrystalline stability has been discussed in the literature before. Indeed, the works of Schuh et al. were already mentioned in our manuscript. The ones of Boyce et al. have been added now in the revised version. We thank the reviewer for bringing them to our attention.

The crucial point is that we go beyond those previous works and provide, for the first time, a thermodynamic treatment of both the bulk and the grain boundary of W-Ti with ab-initio calculations. The previous works were either of conceptual nature (e.g. <https://doi.org/10.1016/j.actamat.2016.12.036> which deals with immiscible NC alloys in general without trying to make statements for specific alloying systems), were dealing with other alloying systems (e.g. <https://doi.org/10.1039/D0NR07180C> was dealing with Pt-Au) or used simplified approaches for alloy thermodynamics (i.e. T. Chookajorn, H. A. Murdoch, and Christopher A. Schuh. Science 337 (2012) 951)._The reviewer will appreciate the shortcomings of previous works when just considering

the mixing enthalpy, which we include here in Figure R1 for convenience. As one can see, the mixing energy is qualitatively different.

Figure R1: Comparison of the mixing enthalpy used in the regular solution model (blue, T. Chookajorn, H. A. Murdoch, and Christopher A. Schuh. *Science* 337 (2012) 951) and the one obtained with ab-initio calculations (black dots, Bodlos et al. ¹).

Now, why have previous assessments been that wrong? The reason is that the simplest way to obtain a miscibility gap in a thermodynamic model is to apply a regular solution model with a positive energy of mixing. However, ab-initio simulations show that miscibility gaps also do arise with negative mixing energy, but one has to depart from the too simplified regular solution assumption. Similar shortcomings can be pointed out for the grain boundary. As shown in the present work, segregation of Ti to W is only expected at high concentrations, while the approach of Miedema that is used by Schuh et al. ²⁻⁵ does not reflect this. Therefore, while these previous works have provided substantial contribution to the conceptual understanding of nanocrystalline stability, their application to specific material systems, such as e.g. W-Ti, need reconsideration.

It is not the scope of the present paper to go into the details of computational thermodynamics. We are currently preparing another manuscript where these aspects will be elucidated in detail. In the present work we provide new experimental and computational evidence on segregation in W-Ti.

We therefore revised the following sentences:

Old: Previous attempts to explain segregation behaviour in W_xTi_{1-x} based on Monte Carlo simulations, have been made on the basis of an idealized thermodynamic description of the W-Ti alloy. However, atomistic calculations of the mixing enthalpy of W-Ti and of the segregation energy of Ti in W cast doubt on the validity of the used parameters which were derived from simplified thermodynamic databases and the Miedema model ⁵. The dependence of the enrichment behaviour with the minority element concentration and its impact on the generated stress profile requires further perception. Such further insights are crucial to fabricate stable nanocrystalline thin films suitable for engineering applications.

New: Previous attempts to explain segregation behaviour in W_xTi_{1-x} have been made on the basis of an idealized thermodynamic description of the W-Ti alloy ²⁻⁵. However, recent atomistic calculations of the mixing enthalpy of W-Ti ¹ differ qualitatively from the previously assumed shape within the regular nanocrystalline solution model. Furthermore, also the segregation energy of Ti in W, as obtained from the

Miedema model⁵ is in qualitative disagreement with results from ab-initio simulations⁶⁻⁹. Therefore, the mechanisms behind stable nanocrystalline thin films are still obscure. Such further insights are crucial to fabricate stable nanocrystalline thin films suitable for engineering applications

Q3. I find the use of vocabulary "incorporating machine learning assisted" really vague. It would be like saying we solve a data-driven problem using computer programming. I would strongly suggest that the author revise their manuscript and be more precise and to the point than saying "machine-learning" as it does not convey ANY useful information to the reader AND gives the impression that the authors are using it as a black box tool not necessarily knowing/understanding what they get out of it.

Thank you for your comment. We rephrased the associated sentences and deleted the phrase "incorporating machine learning assisted" in the abstract and introduction. We specified the phrase ML. We add in the revised manuscript more details with respect to the used ML model.

Q4. I find the description the ML model confusing. The authors mix details about the experimental set up (lines 79-88), some information on so called "trained neural net" (lines 88-102) and then back to some experimental consideration (lines 103-105). I would encourage the authors to rethink this section. This goes back to my comment "3" about the lack of precision in stating "machine learning". The authors seem to be interested in a supervised learning problem with labels. I would clearly state: what is the input, what is the output, what is the intended task at play, and what are the different component used to perform that task. It is unclear reading the subsection how PCA, HOG, SEM and FEM data are all linked together...As for the original draft of the paper, the write up does not give confidence that the authors know how to properly use this information.

Thank you for the comment. We revised the manuscript accordingly. We rearranged the structure of the manuscript to provide a better understanding about how the used methods are linked together. We moved the microstructure investigations in front. Then generated a chapter dealing with the extraction of the deflection data using the GRNN model. Here, we modified the text with respect to the training, input and output specifications as well as provide details about the postprocessing of the gained SEM image data. Subsequently we introduce the stress evaluation using the deflection data. Please see the corresponding changes in this context highlighted in yellow.

Q5. On the FEM model. What not used the closed form "expert model" from the work from Eric Chason. In other words, the authors are basically looking at a multi-fidelity approach (combining experimental data [high-fidelity] with low fidelity data [FEM]) to infer correlations. See comment 1. If anisotropy and other factors are required from the FEM that is not available otherwise then the authors should clearly contrast their approach with known models.

Thank you for the comment. E. Chason's expert model is used to evaluate stresses during the fabrication process by applying in-situ wafer curvature method. The used in situ wafer curvature method provides an average global stress value for each added deposited layer on top. With our ILR method we gain information about the local residual stresses within the thin film stack with a resolution of 15 nm (defined by the cutting step interval) after the single film or multi-layered thin film system on top of a substrate has been deposited and deposition has been concluded. Our developed

FEM model simulates specifically the experimental ILR workflow. Please see also details in the Supplementary. That said it is a different approach and provides possibilities to characterize the formation of the residual stress condition within the thin film after deposition, localize the max. stress value within the layer, and show the actual stress distribution with 15 nm resolution. The approach provides possibility to determine the stress after the deposition. Further impact of curing and life-time cycling performance can be studied.

Q6. Please check spelling (see for example line 150 "previously". There are more throughout the manuscript.

Thank you for the comment. We revised the manuscript accordingly.

Q7. The microstructure analysis could be interesting, but without any mention of the PVD conditions that lead to such texture and grain morphology, it is not that useful. It is well known that depending on the background gas pressure, sputtering power of the targets used, working distance we could have (i) very different microstructures and (ii) very different residual stress states. I am assuming here that the authors work at fixed conditions? Are their changes in the sputtering power for the different at. % of Ti?

We modified the text accordingly and added further information in this context.

Q8. The observations of GB segregation is consistent with computational model (for instance phase-field, see work by Abdeljawad and co-workers). I would recommend that the authors make a note of that and link their observation with known GB segregation mechanisms. Especially if they observed graded compositions.

We thank the reviewer for the very constructive comment. We have added the two references after "A possible approach to stabilize the nanocrystalline structure is by alloying with a minority element^{5,10}"

Furthermore, we have added after

"To understand the reason for this behaviour we focus on the formation energy of bcc W_{1-x}Ti_x in bulk and at the GB"

the following:

"Note that such a diagram has been sketched before in the context of NC stability¹¹ but was not predicted based on ab-initio simulations.

Q9. Can you please add labels of the GB studied in Fig. 4? Also it is unclear if only the W sublattice is shown or else.

Labels for the crystal directions are added as well as a legend for the depicted atoms. As we are considering a bcc GB, the only sublattice present is the W sublattice.

Q10. On the GB segregation. Could the authors comment on the relevance of their simulation results as compared to experimental results especially since deposited GB are most likely not in their ground

state (as it is the case in the simulation) and therefore segregation energetic landscape would differ substantially for defected GB as compared to ground-state one (see for instance work by Schuh and coworkers).

We thank the reviewer for this valuable comment. As mentioned by the reviewer, deposited GBs are most probably not in their ground state structure but rather an excited state that is characterized by the presence of defects like W interstitial atoms, vacancies, disconnections, etc. This excited GB state corresponds to a higher GB energy. The GB segregation energies are closely related to the GB energy, i.e. the maximum segregation tendency of an element increases with the energy of the studied GB but reaches a saturation for high energy GBs¹². Therefore, the expected change in the segregation energy landscape is not a complete shift of the segregation profile but rather a more continuous distribution of segregation energies due to the additional defects in the excited GB structure. A similar conclusion can be drawn from the recent work by Schuh et al.¹³, where the effect of triple junctions and quadrupole nodes of the GB network on the segregation spectrum is studied. In a simple approximation, triple junctions could be considered to be an extremely defected GB, but as shown in the work by Schuh and co-workers, this only slightly affects the segregation energy distribution with an increase in the mean by only 10%.

We include this results in the revised manuscript as:

“Although the real monocrystalline coating contains many different grain boundary orientations which may exhibit relative differences in GB segregation state, we expect that the general segregation trend reported below remains the same] because detailed analysis of segregation at various microstructure defects shows only small changes in the distribution of segregation energies^{12,13}.”

Q11. Overall the whole analysis of using "machine-learning" to correlate residual stress with composition fall way short of the stated objective. The paper only has a very short section on the "ML approach" and the rest of the analysis is rather a classic experimental/modeling analysis of the collected data.

We thank the referee for this valuable comment. First, we added more details with respect to the machine learning-based approach used to extract the deflection from the SEM-image data. Further, your comment provided new ideas with respect to the use of ML. In particular we developed a ML-based feed-forward model consisting of a two-layer multivariable regression architecture to predict the residual stresses for different Ti at%. The model enables us to predict with good accuracy the crossing point from compressive to tensile stress with respect to the minor element concentration. The findings are in good agreement with literature values and fit well to the results obtained from the DFT. Please see the extension in the revised manuscript.

Reviewer #2 (Remarks to the Author):

The manuscript presented an interesting study on the evolution of residual stress in W-Ti films. Ti solutes exist in supersaturated solid solution for 10 and 20%Ti, whereas Ti solutes segregate to grain boundaries. In parallel the residual stress in the films change signs and varies drastically. Extensive

EBSD studies have been performed. While these experimental studies and modeling add something new to the literature, the following concerns must be addressed.

We thank Reviewer 2 for her/his questions and remarks and finding the study interesting. In the following, we address his/her concerns.

Q1. The authors contended that nanocrystalline metals have poor thermal stability, while this is probably true for many cases, such a phenomenon does not constitute a good motivation for the study of residual stress in W-Ti films. For instance, there is no attempt made to investigate the thermal stability of their films. Hence the introduction section should be rewritten to fit in the scope of the manuscript.

Thank you for the comment. We revised the introduction as suggested.

Q2. Given the fact that the sign of residual stress switched from compression to tension between 20 and 30% of Ti, it is unnecessary to investigate a film with intermediate chemistry (25%) to probe the evolution of residual stress.

We think the referee would like to have further information about the minor element concentration where the crossing from compressive to tensile occurs. In order to answer this question, we developed a ML-based feed-forward model consisting of a two-layer multivariable regression architecture to predict the residual stresses for different Ti at%. The ML-based model by using the microstructure information as well as the experimentally obtained local residual stresses along the thin film is capable to estimate the crossing point. The predicted trend shows a non-linear increase. It correlates well with literature values. It also correlates well with the results gained from the DFT calculations.

Q3. Although cantilever beams have been used to quantify residual stress, other methods, such as x-ray diffraction and curvature measurements on conventional (flat) Si substrates have been routinely used to measure residual stress, and one of these methods should be used in parallel to validate their measurement.

It seems that the former submitted manuscript text was not clear enough. We presented in the submitted version for the validation of the ILR results, XRD data. Please see the comparison in table 2 but also a text in the discussion.

New text:

For a validation of the obtained ILR results, we compare the evaluated mean value with results obtained from X-ray diffraction (XRD), see Tab. 1. The XRD, using the $\sin^2 \Psi$ method, provides an integral method to determine global residual stresses in $W_{1-x}Ti_x$ thin films. The mean residual stress determined from the ILR approach for 15 at% Ti and 20 at% Ti provides comparable values to the XRD result. However, for 30 at% Ti a deviation between the ILR mean residual stress and the XRD measurement with about 800 MPa is obtained.

Please see in the discussion the following text:

For 15 and 20 at% Ti the obtained ILR mean stress, as shown in table 1, corresponds well with the XRD results. The observed difference in the mean stress for 30 at% might be linked to the comparatively inhomogeneous lamella distribution illustrated in Fig.2a. The ILR-method, using the micro-beam setup, records locally with high resolution the stress distribution, whereas XRD on the other hand displays an integral method collecting an averaged stress value over a defined surface area, here with 1 x 1cm².

Q4. It is unusual to put discussion and conclusions together. The conclusions must be put into a separate section and serve as a concise summary of major findings.

Thank you for the comment and we modified the manuscript accordingly.

5. There is no detail on film deposition: deposition method, vacuum of the instrument, substrate temperature etc.

We added more information as well as put a reference for more details with respect to the fabrication.

6. STEM has been shown, but TEM evidence is lacking. The diffraction of TEM micrographs should also be provided and indexed.

Thank you very much for the valuable comment. We modified the text in the manuscript and added orientation mappings in the supplementary.

In addition, orientation maps, see supplementary Figure 10, are collected as a useful complement to the TEM contrast images presented in Fig 4 a and e. Hence, we gain further information about the observed fine structure and confirm the polycrystalline structure of the 20 and 30 at% Ti thin film. Different grey values as seen in the TEM contrast images, see Fig 4a and e, indicate the separation of the grain boundaries with differently oriented crystal orientations.

7. Abstract: "We inquire why the experimental observed Ti enrichment can be strongly reduced at smaller concentrations and discuss how it significantly effects the stress stored in the nanocrystalline thin film." Effect is a noun, not a verb.

Thank you for pointing this out. Now ..."affects"...

Reviewer #3 (Remarks to the Author):

In the manuscript "Machine-learning assisted local stress evaluation and atomistic modelling to inquire Ti-enrichment in W-Ti nanocrystalline films", the authors presented a framework containing experimental, numerical and ML methods to study the W-Ti alloy thin films. The residual stress, microstructure, elemental distribution and segregation behavior are clearly studied. The paper is novel and well written. I recommend publication with some revisions.

We thank Reviewer 3 for her/his questions and remarks and finding the paper novel and well written. In the following, we will address her/his questions.

Q1. The results contain too much detail descriptions of simulation methods, which can be moved into the Methods.

Thank you for your comment. We moved the simulation details in the method section.

2. The lamella density and grain density need to be transformed to size, for better comparison with other studies. Besides, what are the lamella structures at microscale? Their density is much higher than that of grain density, and also does not match with the columnar density observed in Figure 3 (20-30 nm for 30% sample corresponding to 1000-2500 μm^{-2}).

Thank you for your comment. The size-based analysis of the lamella density cannot be compared quantitatively with the grain density since the lamellas are not the grains. A grain is rather defined by several lamellas. We try to make this now clearer by adding a figure 1d showing the projection of the grey value FESEM image on top of the FESEM-EBSD data. A high lamella density per grain as shown for 20 and 15 at% is associated with the emergence of compressive stresses in the nanocrystalline films. A small lamella density per grain as shown for 30 at% provides tensile stress.

Q3. Machine learning has become a powerful tool to study the structures and properties of materials. However, it seems that ML is not irreplaceable in this paper. ML is employed to extract the micro-cantilever deflection, which actually can be obtained in many ways. Labeling training images can also be time-consuming, reducing the efficiency of ML. I suggest the authors to directly comparing the accuracy and efficiency of ML framework (including data preparation and model training) with human-annotated methods.

We added the following sentences in the manuscript to address the reviewer's comment:

The root-mean-square error (RMSE) provides for the developed model an inaccuracy of about 2 nm. For a conventional image analysis (Supp. Note 3 and Fig.2) the RMSE is a factor three larger. The evaluated R-squared shows, a model accuracy of 99 %, i.e. demonstrates no underfitting or overfitting. Furthermore, the ML-based approach for the extraction of the deflection in comparison to conventional image analysis saves about 75% in time.

The accuracy, RMSE has been also determined for the second model used to predict the residual stress between 0 and 30 at% Ti.

4. To the best of our knowledge, training a ML model with images often requires a large training dataset. How to confirm that the ML model is well trained without overfitting or underfitting? The predictive error (root mean square error and R2) as a function of training data size should be supplemented.

Please see also the answer to the question above with respect to the newly added paragraph in the revised manuscript.

Q5. A simple $\Sigma 3$ grain boundary was selected for DFT calculations. It is understandably owing to the intrinsic limitations in DFT. But it is well known that the property of grain boundaries can be

significantly influenced by their structures like misorientation angle. The authors could calculate segregation energies of some other grain boundaries with low Σ values. On the other hand, a rough statistic for the misorientation angle distributions in the samples can be helpful to determine the GBs to be calculated.

The GB segregation energies are closely related to the GB energy, i.e. the maximum segregation tendency of an element increases with the energy of the studied GB but reaches a saturation for high energy GBs¹². Therefore, we can assume that the Sigma 3 GB is representative of high angle GBs (as also shown in numerous other studies) and corresponds to the observed orientations in the deposited films (to be checked). This GB or similar high angle GBs will be present in any random microstructures and show most pronounced segregation. The GB segregation energies are closely related to the GB energy, i.e. the maximum segregation tendency of an element increases with the energy of the studied GB but reaches a saturation for high energy GBs¹². Therefore, we can assume that for some more text along that lines. Let us also note here that our treatment of GB segregation goes far beyond previous works in terms of model accuracy. Schuh et al.^{3,5}, for instance, made a crude approximation assuming only a single segregation site which, in addition, was treated with the Miedema model and did not take solute-solute interactions properly into account. We agree that other grain boundaries might show some deviation to the Sigma 3 GB but we are convinced that the essential elements of segregation (which in particular also includes an explanation why segregation is only observed at higher Ti contents in contrast to the model of Schuh) are now captured correctly with our model.

In Figure 3, a thick layer of segregation occurred at the boundary (few nanometers), but in the DFT simulation only 2-3 atomic layers is considered. The reason for the different thickness DFT simulation and in Experiment is the limitation in resolution for TEM. While in DFT, we have single atom resolution, TEM can only resolve about 2 nm small regions. Therefore, the TEM images suggest a broad enrichment, which is not entirely correct. In contrast, HR-TEM or specialized AP-FIM studies show that the actual segregation state is limited to few atomic layers. ([https://doi.org/10.1016/0921-5093\(91\)90318-H](https://doi.org/10.1016/0921-5093(91)90318-H), <https://doi.org/10.1557/jmr.2016.398>)

Q6. In Figure 3, a thick layer of segregation occurred at the boundary (few nanometers), but in the DFT simulation only 2-3 atomic layers is considered.

The reason for the different thickness DFT simulation and in Experiment is the limitation in resolution for TEM. While in DFT, we have single atom resolution, TEM can only resolve about 2 nm small regions. Therefore, the TEM images suggest a broad enrichment, which is not entirely correct. In contrast, HR-TEM or specialized AP-FIM studies show that the actual segregation state is limited to few atomic layers. ([https://doi.org/10.1016/0921-5093\(91\)90318-H](https://doi.org/10.1016/0921-5093(91)90318-H), <https://doi.org/10.1557/jmr.2016.398>)

References:

1. Bodlos, R. *et al.* Ab initio investigation of the atomic volume, thermal expansion, and formation energy of WTi solid solutions. *Phys. Rev. Mater.* **5**, 1–10 (2021).
2. Chookajorn, T., Murdoch, H. A. & Schuh, C. A. Design of stable nanocrystalline alloys. *Science (80-.)*. **337**, 951–954 (2012).

3. Trelewicz, J. R. & Schuh, C. A. Grain boundary segregation and thermodynamically stable binary nanocrystalline alloys. *Phys. Rev. B - Condens. Matter Mater. Phys.* **79**, 1–13 (2009).
4. Murdoch, H. a. & Schuh, C. a. Estimation of grain boundary segregation enthalpy and its role in stable nanocrystalline alloy design. *J. Mater. Res.* **28**, 2154–2163 (2013).
5. Chookajorn, T. & Schuh, C. A. Nanoscale segregation behavior and high-temperature stability of nanocrystalline W-20 at.% Ti. *Acta Mater.* (2014) doi:10.1016/j.actamat.2014.03.039.
6. Scheiber, D., Pippan, R., Puschnig, P., Ruban, A. & Romaner, L. Ab-initio search for cohesion-enhancing solute elements at grain boundaries in molybdenum and tungsten. *Int. J. Refract. Met. Hard Mater.* **60**, (2016).
7. Wu, X. *et al.* First-principles determination of grain boundary strengthening in tungsten: Dependence on grain boundary structure and metallic radius of solute. *Acta Mater.* **120**, 315–326 (2016).
8. Setyawan, W. & Kurtz, R. J. Grain Boundary Strengthening Properties of Tungsten Alloys. in *Fusion Reactor Materials Program Semiannual Progress Report for Period Ending June 30 82–88* (2012).
9. Li, Z.-W., Kong, X.-S., Liu-Wei, Liu, C.-S. & Fang, Q.-F. Segregation of alloying atoms at a tilt symmetric grain boundary in tungsten and their strengthening and embrittling effects. *Chinese Phys. B* **23**, 106107 (2014).
10. Chookajorn, T., Murdoch, H. A. & Schuh, C. A. Design of stable nanocrystalline alloys. *Science (80-.)*. **337**, 951–954 (2012).
11. Abdeljawad, F. *et al.* Grain boundary segregation in immiscible nanocrystalline alloys. *Acta Mater.* **126**, 528–539 (2017).
12. Dösinger, C., Spitaler, T., Reichmann, A., Scheiber, D. & Romaner, L. Applications of Data Driven Methods in Computational Materials Design. *BHM Berg- und Hüttenmännische Monatshefte* **167**, 29–35 (2022).
13. Tuchinda, N. & Schuh, C. A. Grain size dependencies of intergranular solute segregation in nanocrystalline materials. *Acta Mater.* **226**, 117614 (2022).

13th Jan 23

Dear Dr Brunner,

Your manuscript titled "Correlation of atomistic modelling with local stress and microstructure evaluation to inquire Ti-enrichment in W-Ti nanocrystalline films" has now been seen again by Reviewer 2 and 3, whose comments appear below. Reviewer 1 did not respond to our requests to look at the revised paper (causing the delay, which we apologize for) and we therefore asked one of the other referees to comment on your replies to Reviewer 1. In light of their advice I am delighted to say that we are happy, in principle, to publish a suitably revised version in Communications Materials under the open access CC BY license (Creative Commons Attribution v4.0 International License).

We therefore invite you to edit your manuscript to comply with our journal policies and formatting style in order to maximise the accessibility and therefore the impact of your work.

EDITORIAL REQUESTS

* Your manuscript should comply with our policies and format requirements, detailed in our style and formatting guide (<https://www.nature.com/documents/commsj-phys-style-formatting-guide-accept.pdf>).

* Please edit your manuscript according to the editorial requests in the attached table, and outline revisions made in the right hand column. If you have any questions or concerns about any of our requests, please do not hesitate to contact me. It is important that each request be addressed in order to avoid delays in accepting your manuscript. Please upload the completed table with your manuscript files as a Related Manuscript file.

* The editorial requests table also includes a full list of the files that must be provided upon resubmission. Please upload your files according to this table.

* An updated editorial policy checklist that verifies compliance with all required editorial policies must be completed and uploaded with the revised manuscript. All points on the policy checklist must be addressed; if needed, please revise your manuscript in response to these points. Please note that this form is a dynamic 'smart pdf' and must therefore be downloaded and completed in Adobe Reader. Clicking this link will download a zip file containing the pdf.

OPEN ACCESS

Communications Materials is a fully open access journal. Articles are made freely accessible on publication under a [CC BY](http://creativecommons.org/licenses/by/4.0) license (Creative Commons Attribution 4.0 International License). This license allows maximum dissemination and re-use of open access materials and is preferred by many research funding bodies.

For further information about article processing charges, open access funding, and advice and support from Nature Research, please visit <https://www.nature.com/commsmat/about/open-access>

RESUBMISSION

At acceptance, you will be provided with instructions for completing this CC BY license on behalf of all authors. This grants us the necessary permissions to publish your paper. Additionally, you will be asked to declare that all required third party permissions have been obtained, and to provide billing information in order to pay the article-processing charge (APC).

Please use the following link to submit your revised files:

[link redacted]

We hope to hear from you within two weeks; please let us know if the process may take longer.

Best regards,

Xiaoyan Li, PhD
Editorial Board Member
Communications Materials
orcid.org/0000-0002-2953-9267

REVIEWERS' COMMENTS:

Reviewer #2 (Remarks to the Author):

The revised manuscript is acceptable for publication.

Reviewer #3 (Remarks to the Author):

The paper has been well revised with all my concerns answered.